# Experimental Studies of the Processes of Stepped Torque Loads Parrying by Asynchronous Electric Drives and Interpretation of Experimental Results by Nonlinear Transfer Functions of Drives

Vladimir L. Kodkin * and Alexander A. Baldenkov

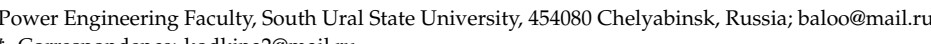

Power Engineering Faculty, South Ural State University, 454080 Chelyabinsk, Russia; baloo@mail.ru
* Correspondence: kodkina2@mail.ru

**Abstract:** This article presents the results of experimental studies of asynchronous motors with a short-circuited rotor (induction motors—IM) with the most common frequency control methods: scalar; vector control; vector with speed control circuit; scalar control with dynamic compensation. The purpose of the experiments was to identify and show the features of the reaction of these control methods to the stepped sketches of the load on the asynchronous electric drive, as in a nonlinear electromechanical system. When selecting the parameters and settings of the "Schneider electric" frequency converter settings, standard instructions and techniques have been used, so the settings of the regulators were not optimal. The aim was not to determine the most effective structures and show fundamentally new solutions. Asynchronous electric motors are substantially nonlinear structures. In the formation of vector control theory, a number of simplifications and assumptions were used, the cost of which was not very clear and theoretically difficult to assess. This work is a step towards estimating the cost of these simplifications. This paper provides the results of experiments in which step-loading modes are presented by the maximum possible registered signals and the amplitude of the stator voltage, formed by various algorithms, the frequency and amplitude of the rotor current and the actual sliding and rotation speed of the engine rotor. This made it possible to maximally objectively assess the effectiveness of the interpretation of asynchronous electric drives and the methods of their regulation. The authors have conducted research in this area for about 15 years. In numerous articles on this subject over the past 25–30 years, the authors did not find such results. The vast majority of work devoted to the study of asynchronous electric drives with frequency control uses the notion of not too significant errors in the linearization of Park's vector equations. In particular, the concepts of the sinusoidality of currents and voltages in AC motors and the accuracy of vector equations are almost always used, which are valid only at a constant frequency of the stator voltage.

**Keywords:** nonlinear systems; electric drive; hyperstability; reactions of electric drives; stepped loads; transfer function of the nonlinear link

## 1. Introduction

A somewhat paradoxical situation has developed in the last twenty years in the frequency control of asynchronous electric drives.

On the one hand, frequency control by asynchronous motors with a short-circuited rotor (induction motors—IM) based on semiconductor frequency and voltage converters (FC) is widespread in industry and power engineering, and FC implement a number of universal control algorithms that ensure application in increasingly complex and precise technological units [1–4]. To date, fundamental unsolved theoretical problems remain, which are associated with significant nonlinearities in the very nature of torque generation in asynchronous electric motors [5–7].

In the most common control methods of asynchronous electric motors, these nonlinearities are not taken into account, moreover, it is not taken into account that vector equations and the resulting substitution schemes and vector diagrams [5–7] describe processes only with sinusoidal currents and voltages of asynchronous motors. When regulating, this is fundamentally incorrect, since in such modes, along with a change in the frequency of the stator voltage, there are changes in the parameters of the replacement circuits: inductive resistances associated with frequencies and the reduced active resistance of the rotor associated with sliding in the IM. Moreover, for vector equations, the base variable changes: the frequency of the stator voltage. That is, to describe the transition from one state to another, it is not just necessary to get new vectors, it is necessary to move from one coordinate system for vectors to another. Vector equations do not take into account these problems at all.

As a rule, standard methods of vector control are constructed on very simplified equations of asynchronous electric motors and take into account completely minor changes, for example, a change in the active resistance of the engine rotor. These changes are insignificant compared, for example, with non-conjuncidality of currents during regulation. But, as a rule, the influence of these drives on the actuator is not estimated.

All studies devoted to vector control methods are based on the decomposition of all variables: rotor and stator currents, voltages, EMF, magnetic fluxes and flux links into longitudinal and transverse components on the "d" and "q" axes. This automatically implies the sinusoidality of all processes. At the same time, all manufacturers of frequency converters offer a huge selection of devices for sale, including chokes and filters, that improve the harmonic composition of the same motor currents. Furtermore, the influence of "extra" harmonic components on the control characteristics and on the accuracy of maintaining and regulating speed or effort is almost never mentioned.

Another very important simplification is the generally accepted assumption of "slowness" of electromagnetic processes in the rotor [6], which is very convenient, but very erroneous when parrying fast loads. At the same time, the theoretical assessment of this error is associated with the analysis of highly nonlinear differential equations and it is practically impossible to make it accurate for engineering calculations. Meanwhile, in practice, engineers face serious problems. Very often the drives of mechanisms of cranes, transport mechanisms and other technological devices, experiencing a variable load, work in a completely different way than expected of them. They have significant "dynamic" errors in speed or effort, lag behind the required pace of movement, or work with large current overloads. It is very difficult to give a theoretical assessment of these problems.

For systems with nonlinearities and complex mathematics, experiments play a very important role, including the parrying of a step load. For electric drives and asynchronous electric drives in particular, this mode is the most difficult. The control error increases significantly precisely at the point of torque loading.

## 2. Procedure for Conducting Experiments

The experiments were carried out on a stand (Figure 1) on which two frequency-controlled asynchronous electric drives operated on a common mechanical shaft, testing the parameters of the motor and frequency converters in various operating modes.

In the working drive (QF2, UZ2, M2), the control mode under study was set to either: scalar control, vector sensorless control, or control with speed feedback. Depending on the the signal supplied to the input of the frequency converter UZ2 of the working motor M2, the drive is output to a certain rotation speed (and the corresponding frequency of the stator voltage). After a certain time interval, a task is sent to the frequency converter UZ1 of the load motor M1. The operating mode of the load drive is determined by the task for the rotation speed.

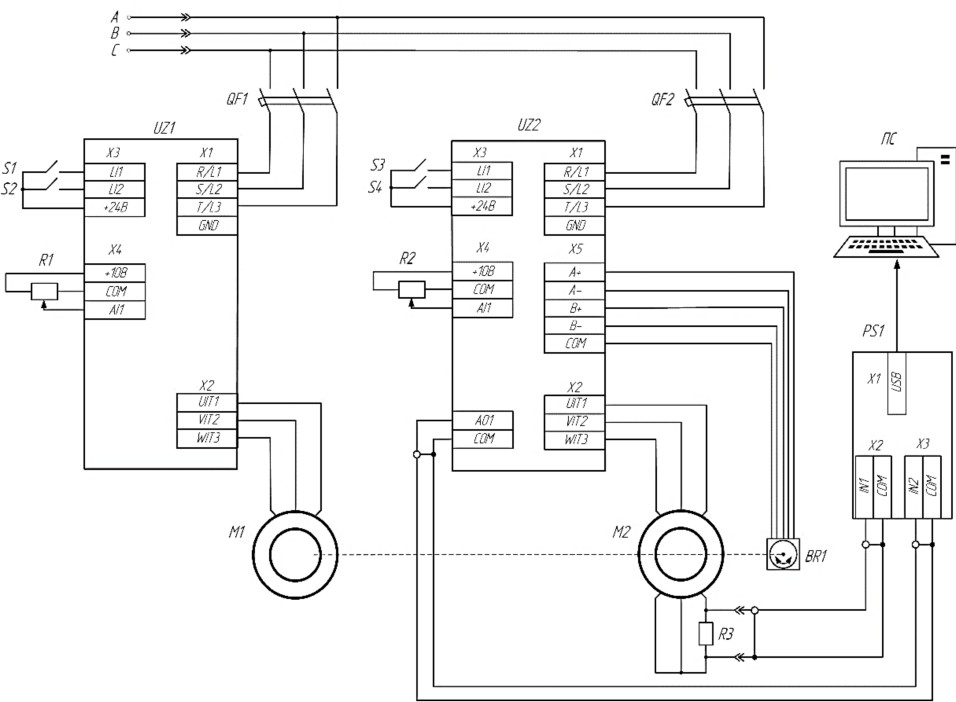

**Figure 1.** Electrical diagram of the stand.

The block diagram of the sensorless control is shown in Figure 2. Corrections under load are formed according to the signals of the stator current and the parameters of the stator voltage. Scalar and vector voltages differ in correction algorithms, which are described in detail in many works [8–12]. According to the stator current and speed diagrams in the experiments, the differences are not very noticeable, Figure 3.

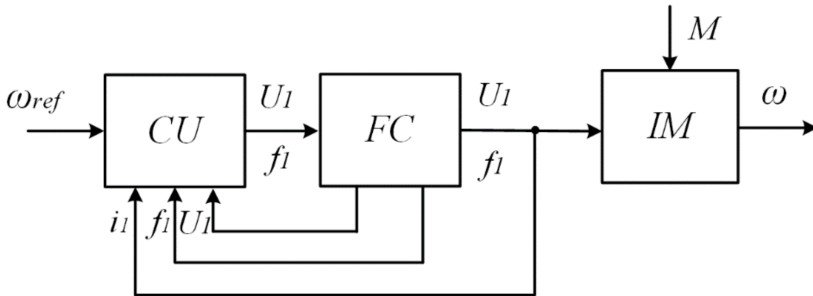

**Figure 2.** Block diagram of an asynchronous electric drive with scalar and vector sensorless control. CU, control unit; FC, frequency converter; IM, induction motor.

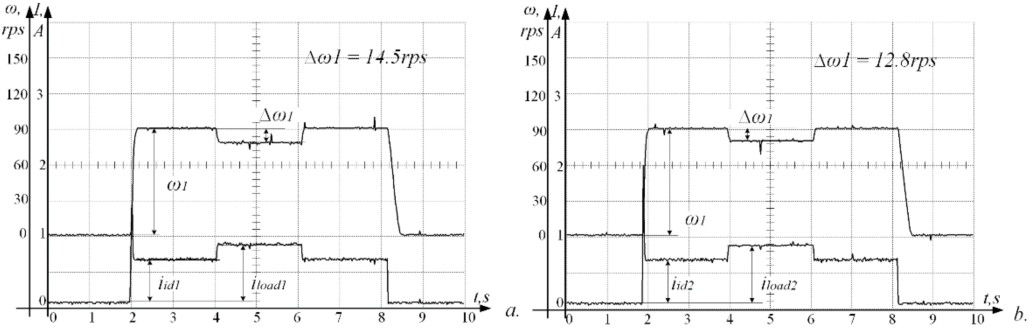

**Figure 3.** Diagrams of processes in asynchronous drive with scalar (**a**) and vector sensorless (**b**) control.

Diagrams of the processes of parrying the load are shown in Figure 3. The experiments are described in detail in a number of papers [13–16].

It should be noted that the parameters for vector sensorless control were selected according to the methods and instructions for ATV series frequency converters and may not be optimal (Figure 3b). It is also known from numerous works that standard vector algorithms do not provide compensation even for the static load of the electric drive in a wide range of rotation speeds and applied loads. A lot of work is devoted to the use of "observers" of such compensation. However, the operation of these devices is not without problems and to date serial frequency converters use them quite rarely. The processes are well interpreted by the mechanical characteristics of the drive—a widespread method that graphically shows how the operating modes in engines change under load.

With a scalar control without compensation (Figure 4), at the operating point $A$, mechanical characteristic $n_1(M)$ is shifted to the zone of large slides: the point $B$. The processes in a system with a rattling vector control $n_v(M)$, $n_2(M)$ will be close to this.

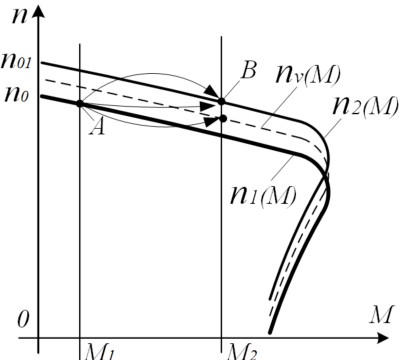

**Figure 4.** Mechanical characteristics of an asynchronous electric drive. $n_1(M)$, mechanical characteristics without compensation with scalar control; $n_v(M)$, with compensation; $n_2(M)$, intermediate version. $A$ is the operating point at minimum load. $B$ is the operating point after loading.

In a system with vector control and a PID-regulator of speed, the processes change significantly (Figures 5 and 6). In a system with speed vector control and PID-regulator, the processes change significantly (Figure 6). In such systems, complete static load compensation is ensured, that is, after the end of the transition process, the speed corresponds to that specified. In this case, the dynamic process of its recovery, as shown in numerous experiments, is practically independent of the parameters of the PID controller: the values of the gain and integration time. As the analysis of assumptions made in vector control showed, the reason for this in the simplification of equations that equate the frequency of the stator voltage with the engine rotation speed. When controlling the drive in the input of the speed task, this assumption does not have a significant effect, but when the load is parry, after the dynamic rate of speed, according to this algorithm, the frequency of the stator voltage increases, which reduces the engine moment and slows down the dynamics.

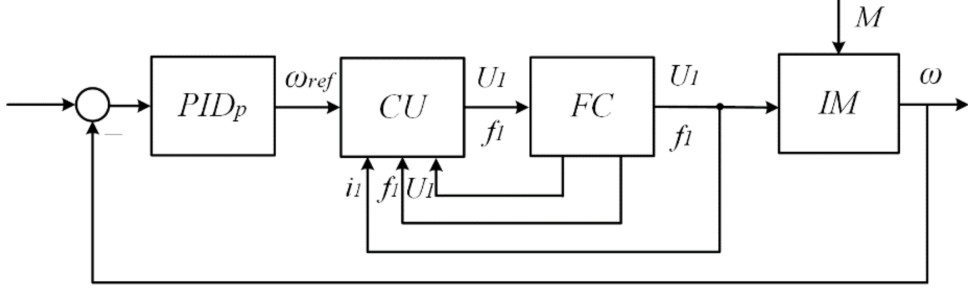

**Figure 5.** Structural scheme of an asynchronous electric drive with vector control and PID-regulator.

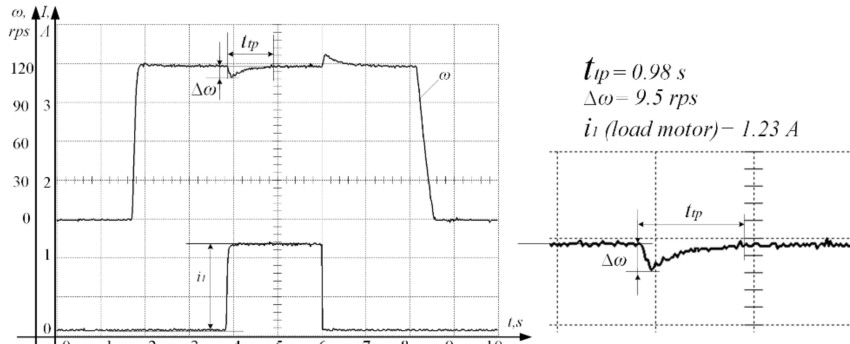

**Figure 6.** Asynchronous drive process charts with speed vector control and PID-regulator.

As shown by the experiments on the stand, the frequency of the stator voltage and the magnitude of the sliding change dramatically. The speed recovery process is very slow, but with a precise value in the steady mode.

The mechanical characteristics of the asynchronous electric motor definitely show the features of a load parry process in which the PID controller and the vector control method select the stator voltage parameters and the mechanical characteristics to compensate for the load. The speed of rotation after the load is restored to the desired value (p. B, Figure 7), but the process, in accordance with the algorithm, passes along the "slow" trajectory.

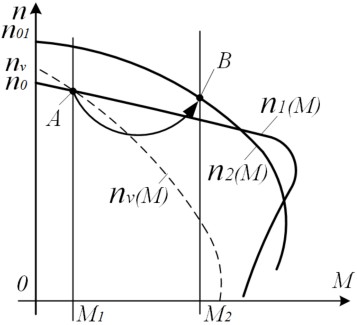

**Figure 7.** Mechanical characteristics of an asynchronous drive with vector control and a PID speed controller. $n_1(M)$ is the initial mechanical characteristic at low load, $n_v(M)$ is the mechanical characteristic with increasing $f$, $n_2(M)$ is the mechanical characteristic of the drive with zero speed error and increased frequency of the stator voltage.

In this paper [14], a correction is proposed, which is called dynamic positive feedback (DPF). The effect of this correction turned out to be very effective, as follows from the diagrams and mechanical characteristics (Figure 8). Unlike well-known load compensation schemes, this correction contains a dynamic link that allows you to provide a "deep" correction without disturbing the system stability. This paper presents experiments showing the mechanism for the influence of this connection to a nonlinear asynchronous electric motor.

It is quite difficult to establish the "physical" reasons for the differences in the processes. In the absence of information about the real, uncorrected frequency of the stator voltage and information about the processes in the motor rotor only from the speed diagrams and the current of the stator, it is practically impossible.

In this regard, a motor with a phase rotor was installed on the test stand in place of the working motor, which allowed the recording of instantaneous values of the rotor current. Considering that the frequency of the rotor current is an exact analogue of absolute slip in the steady modes and in transition processes, it does not make sense in transition and this frequency and sliding, as well as the rotor current, has only an active component, it should be recognized that this information should be considered very well: the effectiveness of the management algorithm under study.

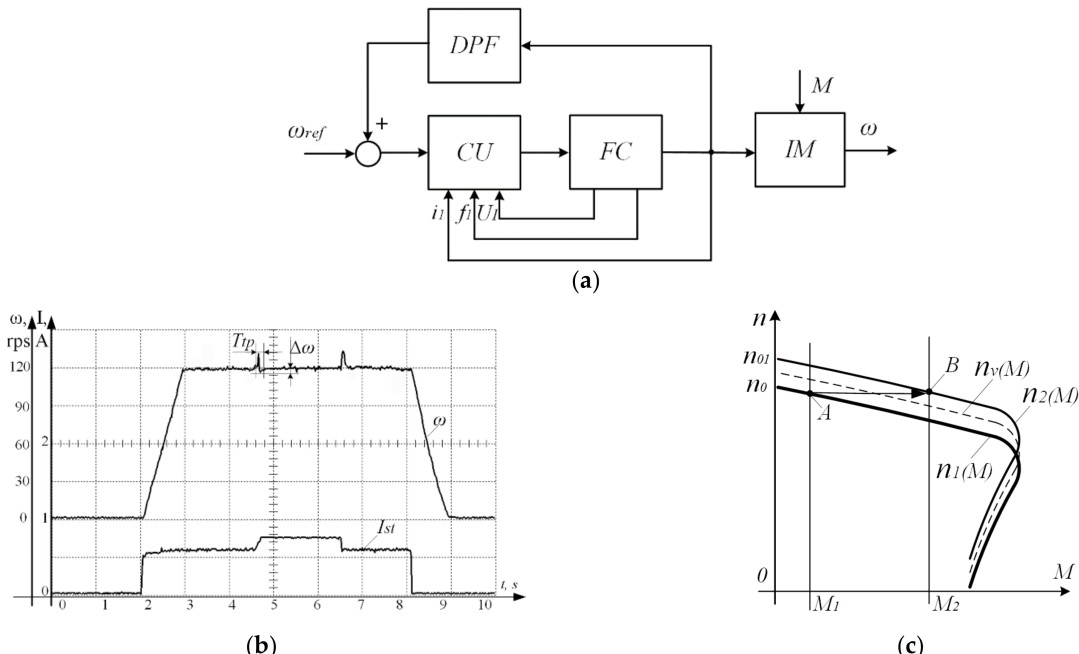

**Figure 8.** Block diagram of an AED with dynamic coupling by torque (**a**), transients (**b**) and mechanical characteristics (**c**). $n_1(M)$ is the initial mechanical characteristic at low load, $n_2(M)$ is the mechanical characteristic corrected by DPF at increased voltage on the stator.

Figures 9–11 show diagrams of rotor currents with three control methods. After the end of transient processes, the frequencies of rotary currents are constant and determined quite accurately, as are vector control and dynamic compensation. The frequency of the main harmonic of the rotor current in the drive with DPF (3.5 Hz) is significantly lower than in the drive with a PID speed controller (8.125 Hz). The amplitudes also differ significantly: 1.1 A versus 2 A (Figures 9 and 10).

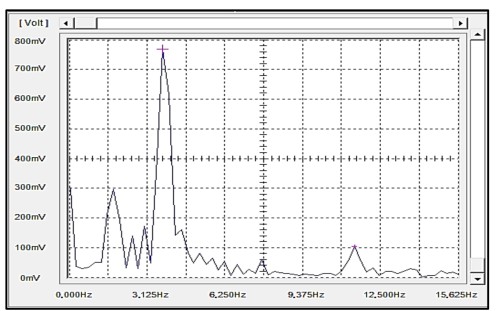 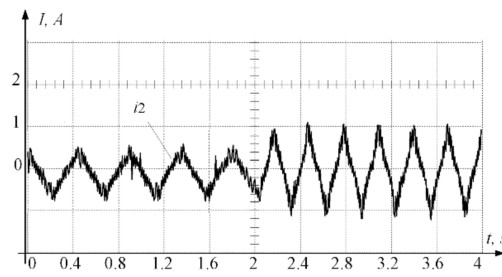

**Figure 9.** Rotor current in a DPF drive and its spectrum under load surge.

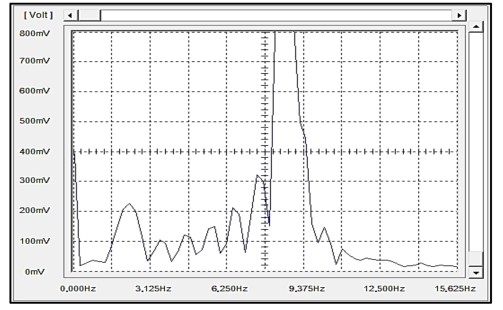 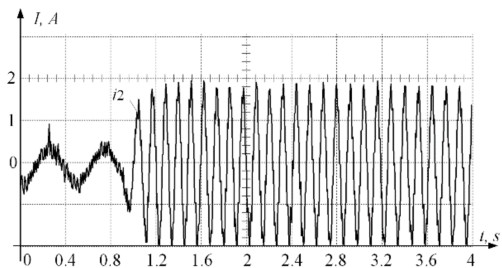

**Figure 10.** Rotor currents and current spectrum in the drive with vector control and speed feedback.

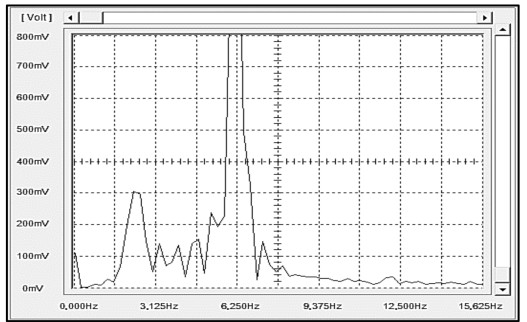 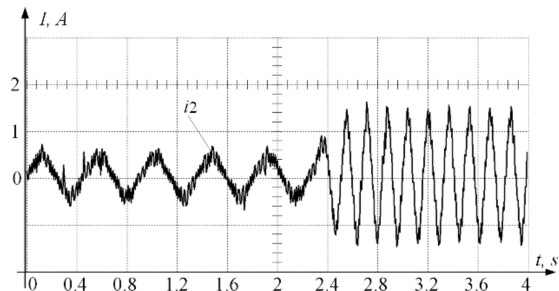

**Figure 11.** Rotor currents and current spectrum in a scalar controlled drive.

It was not possible to explain all the features of the processes or, most importantly, to suggest the most effective ways of correction, while remaining within the framework of "vector" theories.

To overcome these problems, instead of vector equations [8–10], a formula for the nonlinear transfer function of the link forming the torque was proposed, which can be called the nonlinear transfer function or the Kloss dynamic formula:

$$W(p) = \frac{2M_k(T_2'p+1)S_k}{\omega_1[(1+T_2'p)^2 S_k^2 + \beta^2]} \tag{1}$$

there, $\omega_1$ is the frequency of stator voltage and $\beta$ is the relative slip, depending on the drive load.

The structural diagram of the IM with the FC, which forms the torque of the $M_k$, will take the form shown in Figure 12 (the transfer function does not take into account the pulse character of the FC).

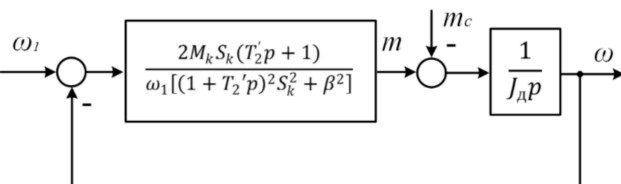

**Figure 12.** Structural diagram of IM with nonlinear link of torque forming.

The nonlinear transfer function corresponds to the frequency characteristics family, varying with the same parameters. The identification of drives with IM using these categories is described in detail in a number of articles [14–16]. The transfer function (TF) and the frequency characteristics (FC) determine the dependence of the dynamics of asynchronous drives from the frequency of stator voltage and the relative slip, as described in [14].

According to its physical (or mathematical) content, this formula is the interpretation of a complex nonlinear mathematical operation with a Laplace transformation with a linear component, the individual parameters of which may vary, while maintaining the continuity of mathematical expression (as opposed to vector equations).

## 3. Changing the Transfer Function

Consider another simplified equation that will allow the evaluation of the effect on transient processes during stepped load surges of the nonlinear transfer function.

Consider a nonlinear structure with a transfer function $W(p, Y)$, depending on y (Figure 13).

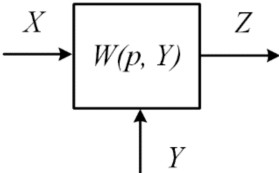

**Figure 13.** Scheme of a nonlinear structure.

Let the transition equation in this nonlinear structure in increments ($\Delta Z$) look like this:

$$\Delta Z = \Delta X \cdot W_1(p, Y) + X \cdot \Delta W(p, Y) = \Delta X \left[ W_1(p, Y) + \frac{\Delta W(p, Y)}{\Delta X} \cdot X \right]$$

For step load $X = \Delta X$, then:

$$\begin{aligned}
\Delta Z &= \Delta X [W_1(p, Y) + \Delta W(p, Y)] \\
\Delta W(p, Y) &= W_2(p, Y) - W_1(p, Y) \\
\Delta Z &= \Delta X [W_1(p, Y) + W_2(p, Y) - W_1(p, Y)] = \Delta X \cdot W_2(p, Y)
\end{aligned} \tag{2}$$

where $W_1(p, Y)$ and $W_2(p, Y)$ are the transfer function of the nonlinear link at the beginning and end of the transition process at $\Delta X = X$ and changes in the transfer function $W$ from the change $Y$. That is, the reaction of a nonlinear link with a changing transfer function is determined by the final value of the transfer function. This determines the following: for a structure with a nonlinear transfer function, the result of the transient process will be not only a transient process in an electric drive, but also a transfer function of a nonlinear structure formed as a result of the process.

For the nonlinear transfer function of an asynchronous electric motor, the determining values, in addition to the frequency of the stator voltage $\omega_1$ and sliding $\beta$, will be the amplitude of the stator voltage $U_1$ and its frequency $f$, which determine the critical torque, $M_k$, which is proportional to the square of the ratio $U/f$. These parameters, along with the values of the stator and rotor currents and the motor speed, identify the motor control method.

### 4. Experimental Studies of the Reactions of Electric Drives to Stepped Loads

Experiments have been carried out to evaluate the changes in transfer functions with stepwise load changes.

The nonlinear transfer function (1), as mentioned above, allowed for a correction with high efficiency, which made it possible to assume sufficient accuracy of the proposed interpretation. More detailed research was conducted.

The working drive in which 5 control methods are sequentially implemented, "manages" the task for rotation speed corresponding to a given frequency—**20 Hz**.

The load drive generates a "counter" moment at a certain point in time corresponding to the mechanical characteristic with a given rotation frequency of **15 Hz**.

The parameters of the currents in the rotor and stator of the working drive in the initial mode (without load) and in the steady state after the counter activation of the load drive are given in Tables 1 and 2.

**Table 1.** At low load, the load drive is switched off.

|  | $f_1$, Hz | $U_1$, V | fs, Hz | $I_1$, A | $f_2$, Hz | $I_2$, A | $f_{1a}$ | $U_1/f_1$ |
|---|---|---|---|---|---|---|---|---|
| SC | 20 | 91 | 18.6 | 0.5 | 1.5 | 1.48 | 20.1 | 4.5 |
| VC | 20 | 83 | 18.1 | 0.4 | 1.75 | 1.88 | 19.85 | 4.3 |
| SVC | 20 | 90 | 20 | 0.39 | 1.72 | 1.76 | 21.76 | 4.3 |
| DPF | 20 | 92 | 18.6 | 0.42 | 1.42 | 1.3 | 20 | 4.5 |
| DPF2 | 20 | 103 | **19.7** | 0.36 | **1.3** | **1.3** | 21 | **4.9** |

**Table 2.** At a load of 70% of *Mm* (the load is equivalent to the load drive setting of 15 Hz).

|      | $f_1$, Hz | $U_1$, V | fs, Hz | $I_1$, A | $f_2$, Hz | $I_2$, A | $f_{1a}$ | $U_1/f_1$ |
|------|-----------|----------|--------|----------|-----------|----------|----------|-----------|
| SC   | 20        | 91       | 6.1    | 0.7      | 13.8      | 12.8     | 19.9     | 4.5       |
| VC   | 20        | 84       | 0      | 0.8      | 20        | 15.8     | 20       | 4.2       |
| SVC  | 20        | 111      | 11.8   | 0.7      | 14.9      | 14.6     | 26.7     | 4.2       |
| DPF  | 20        | 103      | 10.6   | 0.7      | 12.2      | 11.3     | 22.8     | 4.6       |
| DPF2 | 20        | 110      | **12.9** | 0.7    | **9.6**   | **10**   | 22.5     | **4.9**   |

The values of the voltage $U_1$, the amplitude of the stator current $I_1$, the actual rotation speed *fs* and the recalculated FC relative to the specified frequency $f_1$, were recorded according to the readings of the FC monitor and the rotor current $I_2$ and its frequency $f_2$ by the Instek GDS-2062 oscilloscope.

According to the values of the measured rotational speed $f_s$ and frequency of the rotor current $f_2$, the actual values of the frequency of the stator voltage $f_{1a} = f_2 + fs$ and the ratio $U_1/f_1$ were calculated, which determine, as indicated in [15,16], the main magnetic flux in the motor.

The frequency of the rotor currents $f_2$ determines the actual slip in the working drive.

Experiments were carried out with the operation of the DPF compensation described earlier, both with standard parameters and with the operation of this compensation with a significantly increased feedback link gain (DPF2).

### 4.1. Comment 1

First, let us analyze the operation of the drive from the traditional point of view—by the rotor stator currents and speed errors. In these experiments, the ratio in each case does not change under load (a feature of the frequency converter). The working point on the mechanical characteristic shifts along a parallel curve, as in Figure 8c. The stator current changes little with increasing load and torque of the "working" electric motor, 0.5 A without load and 0.6—0.7 A with loads; its changes cannot be analyzed.

The rotor current varies significantly, from 1.3 to 1.88 A without load and from 10 to 16 A at load.

It is obvious that it is the rotary current, being active, that creates the torque.

The control algorithms DPF and DPF2 rely on a continuous nonlinear transfer function which is a more accurate interpretation of the AED since it does not have the assumptions that are made in the SVC (only the main harmonics in the currents and EMF of the motor, $\omega_1 = \omega$, $\frac{d\varphi_2}{dt} = 0$), "choose" control more efficiently.

This correction method is certainly more promising both for static and quasi-static modes and for operation under complex disturbances and in complex technological systems (special vehicles, wind turbines, UAVs, technological complexes, power engineering, etc.).

Now, let us consider the transfer functions of the electric motor torque generation channel, determined by the control method and the load. In accordance with the formula, the TF and FC are determined by the critical torque $M_k$, the critical slip $s_k$ and the current slip $\beta$ determined by the load (Figure 14).

It is necessary to pay attention to the fact that without load, the frequency characteristics of the drive differ little, only the electric drive with DPF2 has increased coefficients when static and at low frequencies. This is reflected in lower stator and rotor currents as well as lower absolute slip.

Under load, there is a significant change in the frequency characteristics both at low frequencies and in the mid-frequency zone. This is especially important in a drive with a PID controller. This explains both the large stator and rotor currents and the high frequency of the rotor current, that is, a significant absolute slip.

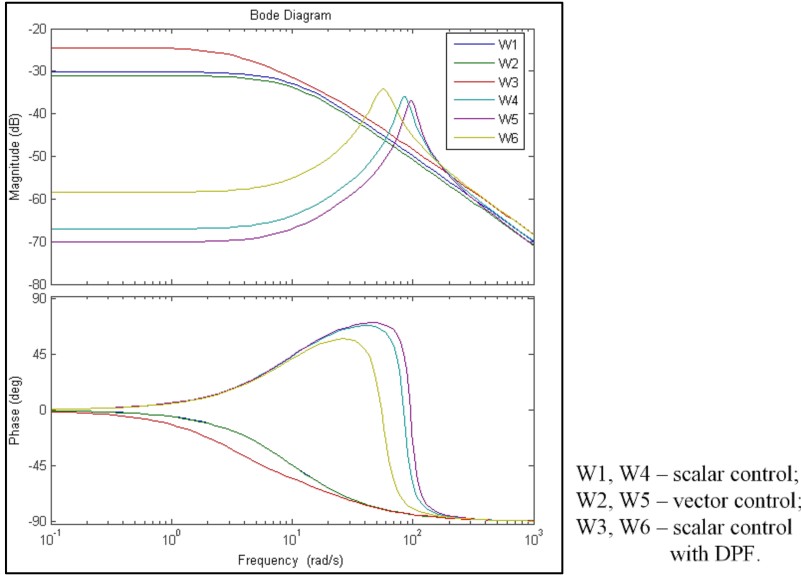

**Figure 14.** Frequency response and phase response asynchronous drive with different control algorithms (W1, W2, W3—with low load, W4, W5, W6—when load surges).

In the version with DPF2, the transfer function and frequency characteristics of the engine change significantly: with DPF the parameters improve and with DPF2 they are practically linearized, eliminating potential oscillation, which can be estimated from the phase response.

That is, the interpretation of the transfer function turns out to be quite accurate and the system can be called hyperdynamic, that is, changing in structure.

The following experiments were carried out only for the DPF2 variant with increased motor flows with an increasing amplitude of the stator voltage. The drive was set to a rotation speed corresponding to a frequency of 30 Hz, Table 3 shows the data when operating with a small load.

**Table 3.** At low load, the load drive is switched off.

|  | $f_1$, Hz | $U_1$, V | $fs$, Hz | $I_1$, A | $f_2$, Hz | $I_2$, A | $f_{1a}$ | $U_1/f_1$ |
|---|---|---|---|---|---|---|---|---|
| 1 | 30 | 137 | 28.4 | 0.432 | 1.695 | 1.8 | 30.1 | 4.6 |

Table 4 shows the data of the drive with a load with an increase in the main magnetic flux in the motor.

**Table 4.** At 50% of rated load (load equivalent to 10 Hz load drive reference).

|  | $f_1$, Hz | $U_1$, V | $fs$, Hz | $I_1$, A | $f_2$, Hz | $I_2$, A | $f_{1a}$ | $U_1/f_1$ |
|---|---|---|---|---|---|---|---|---|
| 1 | 30 | 137 | 24 | 0.480 | 5.8 | 5.8 | 29.8 | 4.6 |
| 2 | 30 | 150 | 25.2 | 0.496 | 5 | 5.44 | 30.6 | 5 |
| 3 | 30 | 180 | 26.7 | 0.600 | 3.3 | 5.12 | 30 | 6 |
| 4 | 30 | 200 | 27.4 | 0.680 | 2.63 | 4.48 | 30 | 6.8 |

The DPF connection was corrected in such a way that under load the voltage amplitude increased to 200 V.

### 4.2. Comment 2

With an increase in the ratio $U_1 \backslash f_1$ and the main magnetic flux in the motor, the rotor and stator currents and the absolute slip required to parry the load are reduced.

The frequency characteristics also change significantly.

At light load, the frequency response is close to that of the inertial link (W7). With increasing load, the frequency response is distorted and the structure becomes oscillatory (W8, W9). As follows from the frequency characteristics shown in Figure 15, built according to the experimental data given in Table 4, with an increase in the main magnetic flux, the speed of the torque formation circuit, rigidity and stability margin all increase. The drive with the maximum ratio again becomes a link close to the inertia (W10)

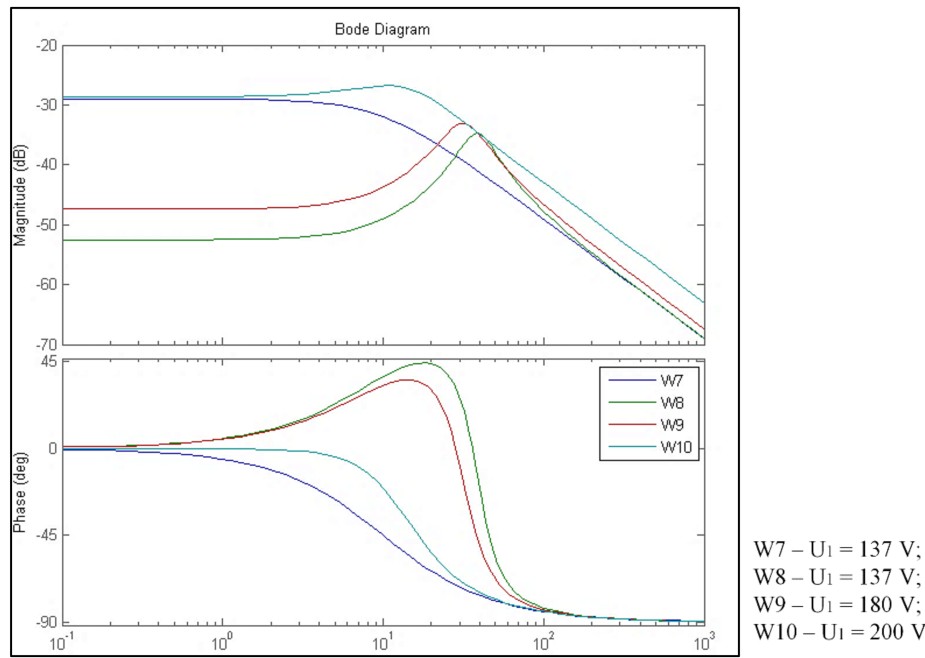

W7 – U₁ = 137 V;
W8 – U₁ = 137 V;
W9 – U₁ = 180 V;
W10 – U₁ = 200 V.

**Figure 15.** Frequency response and phase response of an asynchronous drive with scalar control with DPF2 at various values of stator voltage. W7—at low load; W8, W9, W10—under load.

Previously, in experiments on drives of a production line, the viability of improving the stability of drives with non-rigid connections after the introduction of such a correction was established—a rare result for automatic control systems, especially for nonlinear automatic control systems. The experiments presented in the article confirmed the correctness of the proposed interpretation.

### 5. Discussion of the Results of the Experiments

The processes of parrying stepped loads studied in this article made it possible to evaluate the effectiveness of various algorithms for frequency regulation of asynchronous electric motors. The process of parrying loads in induction motors is a hyperdynamic process in which a nonlinear continuous transfer function is modified and is used to interpret complex mathematical transformations in induction motors that are not described by exact traditional mathematics and known methods of engineering analysis. The "final" transfer function of the electric drive, which determines its dynamic properties, is determined by the selected control algorithms and the main magnetic flux in the motor. The "selection" of the values of the amplitude and frequency of the voltage on the stator after the end of the transients and the change in the transfer functions of the motor depends on the control algorithm: SC, SVC, or DPF. Traditional control algorithms (SC, SVC) choose U and f suboptimally, since the algorithms are based on vector equations and a number of assumptions that are very erroneous for operation under load even in static modes ($\omega=\omega$,



for example). The "final" transfer functions after parrying the loads turned out to be drives with low values of the main magnetic flux and "slow" dynamics. This is confirmed by the timing diagrams of the currents of the rotor speeds obtained experimentally. In these experiments, the trajectories of transitions from one control state to another were chosen inefficiently. One of the significant reasons for this situation is the use of vector equations for asynchronous electric motors which were originally compiled for constant stator voltage frequencies and constant load torques. For dynamic modes of operation, which are "hyperdynamic" for nonlinear electromechanical systems, i.e., changing their structure, these equations are erroneous and the regulation of asynchronous electric drives based on these equations is ineffective.

Continuous nonlinear transfer functions interpreting the AED turned out, as experiments have shown, to be more accurate in describing the processes of parrying load bursts. Correction by local feedbacks, proposed as a result of the analysis of continuous nonlinear transfer functions, turned out to be more effective in load parrying modes. This was shown by the formation of faster and more stable trajectories of transitions from one static state to another. Additionally, the accuracy of maintaining the speed of rotation of asynchronous electric motors under load turned out to be higher. This determines the advantages and prospects for the use of continuous transfer functions and continuous local corrections and structures in the AED of complex technological complexes.

## 6. Conclusions

Variable speed induction drives are one of the most widely used electromechanical systems in today's industry. The same electric drives are an example of systems with nonlinear links. The method proposed in a number of works for describing these nonlinearities—the identification of an electric drive by a system with a variable transfer function, made it possible to overcome a number of problems with traditional vector equations adopted in the description of such electric drives. Parrying step load jumps is one of the most difficult modes of operation of nonlinear electromechanical systems. The traditional mathematical description of processes in asynchronous electric drives by vector equations leads to significant errors when comparing calculations and experimental data. This is due primarily to the fallacy of the methods of analysis of these processes. An important role in improving the methods of controlling asynchronous electric drives, as nonlinear electromechanical systems, is played by experimental research. It was the experiments that proved that the method proposed in the article for identifying nonlinear systems by continuous nonlinear transfer functions and families of frequency characteristics makes it possible to give a qualitative and accurate description of the processes of parrying loads. The experiments also showed that the correction of asynchronous electric drives, developed on the basis of nonlinear transfer functions, is most effective in terms of the smallest slip, as well as the smallest currents of the stator and rotor of the electric motor, which are formed to counter dynamic load surges.

**Author Contributions:** Conceptualization, V.L.K.; methodology, A.A.B.; software, A.A.B.; validation, V.L.K.; formal analysis, V.L.K.; investigation, V.L.K. and A.A.B.; resources, V.L.K.; data curation, V.L.K.; writing—original draft preparation, V.L.K.; writing—review and editing, A.A.B.; visualization, A.A.B.; supervision, A.A.B.; project administration, V.L.K. All authors have read and agreed to the published version of the manuscript.

**Funding:** This research received no external funding.

**Data Availability Statement:** The raw data supporting the conclusions of this article will be made available by the corresponding author upon reasonable request.

**Conflicts of Interest:** The authors declare no conflict of interest.

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
