# Peer review of "Experimental Studies of the Processes of Stepped Torque Loads Parrying by Asynchronous Electric Drives and Interpretation of Experimental Results by Nonlinear Transfer Functions of Drives"

_processes, doi:10.3390/pr10061204_

Round 1
Reviewer 1 Report
The authors have tried to improve manuscript. Now, it has been improved but it still needs some minor improvement.
(1) The authors should enhance introduction since it is less described and it is also short.
(2) References are not properly written. Recheck all references and write all references in same format.
Author Response
Comment in app

Reviewer 2 Report
This article deals with asynchronous electric drives with frequency control, one of the most used electromechanical systems in modern industry. Thus, the same electric drives exemplify systems with nonlinear links. The method proposed in several works to describe these nonlinearities the identification of an electric drive with a variable transfer function made it possible to overcome several problems of traditional vector equations adopted in the description of such electric drives. Parry step load surges are one of the most difficult modes of operation for nonlinear electromechanical systems. Related to this are the methods of analyzing these processes and the importance of experimental studies.
The proposed method for identifying nonlinear systems by continuous nonlinear transfer functions and families of frequency characteristics gives a qualitative description of the load-sharing processes. Correction of asynchronous electric drives, developed based on nonlinear transfer functions, is the most effective in terms of lower slip as well as electric motor stator and rotor currents. The article is well-written and well-grounded, with new contributions. Therefore, it is accepted for publication.
Author Response
Comment in app

Reviewer 3 Report
First of all, I think, the manuscript was poorly prepared by the author. A lot of improvements should be considered before it is accepted for publication.
- Please add a more recent paper or journal to prove the issue is contemporary. On top of that, does reference no 18 relate to this paper?
- The order of reference in the list of references should reflect its appearance order in the text. On top of that, most of the references were not cited in the text. Furthermore, please use the standardized style of writing for reference.
- Please use the correct format such as when referring to a figure (use either Fig. or Figure not both with compliance of the paper guideline). On top of that, do not make a simple mistake like on page 3 of 13, line 85. Should it have a reference in the bracket?
- The explanation of the paper was not easy to follow and understand. The author only briefly mentions the figure, equation, and result without any further explanation. It made the contribution of the paper unclear.
- The claim by the author of doing the research for about 15 years and numerous articles over the past 25 to 30 years should be reflected in the article by explaining the research gap that the author wants to cover. The research gap should be proven by providing adequate references.
- The result of the experiment should be compared with the conventional technique to prove it produces a better outcome.
Author Response
Comment in app

Round 2
Reviewer 3 Report
The explanation of the paper was not easy to follow and understand. The author only briefly mentions the figure, equation, and result without any further explanation. It made the contribution of the paper unclear.
The claim by the author of doing the research for about 15 years and numerous articles over the past 25 to 30 years should be reflected in the article by explaining the research gap that the author wants to cover. The research gap should be proven by providing adequate references.
The result of the experiment should be compared with the conventional technique to prove it produces a better outcome.
This manuscript is a resubmission of an earlier submission. The following is a list of the peer review reports and author responses from that submission.
Round 1
Reviewer 1 Report
This article presents that trimming step load peaks is one of the most difficult operating modes for
linear electromechanical systems. The methods of analysis of these processes and the importance of experimental studies are related to this. So,
the proposed method for identifying nonlinear systems by continuous nonlinear methods
transfers functions and families of frequency characteristics, making it possible to give a
qualitative description of load trimming processes. With all this, the
correction of asynchronous electrical drives, developed based on nonlinear
transfer functions, was the most effective in terms of the lowest slip and the currents
of the stator and rotor of the electric motor. This article is well written and well formulated. Therefore, the article is accepted for publication.
Please see the attachment.

Reviewer 2 Report
General remarks.
- In my opinion, the paper presented falls outside the scope of the MDPI Processes journal and, therefore will not reach the right audience. The article should be sent to MDPI Energis or MDPI Electronics for example
- Of course, typical modern control systems - both scalar controls and especially sensorless vector controls take into account the non-linear phenomena occurring in induction motors. For example, it is typical to use an adaptively changed equivalent value of the rotor resistance. Hence the authors' initial thesis is incorrect.
- The ( not very clear) waveforms in figure 3a are understandable. Since the system operates in an open loop, there is a steady-state speed error when the load torque is applied. However, in the case of vector control, the steady-state error should be reduced to zero. Therefore, in the case of the experiment in Fig. 3 b, were there no errors in the selection of the drive parameters? The waveforms in Figure 6, on the other hand, indicate an ill-tuned PID controller - too low a gain and too high an integration time constant.
- Figure 4 is a well-known, textbook figure.
- Figure 7 is completely wrong. If we assume that we treat the drive and the controller as a whole, the mechanical characteristics are a straight line - in steady state the speed does not depend on the load, within the operating range. If, on the other hand, the characteristics are to concern only the machine, they are identical as for the scalar control, but the frequency and phase of the supply voltage change dynamically.
- The DPF mechanism, proposed as a novelty, is a well-known compensation in an open loop control system
- The frequency measurement in the rotor will only correspond to a slip if the supply frequency is constant. However, since in the case of vector control this frequency varies, the proposed method of measuring slip is wrong.
- The selection of literature is very selective and unrepresentative
Detailed remarks
- It is not clear what the authors mean by 'parrying'.
- The term 'moment' is used throughout the work, instead of the commonly accepted torque. This makes it difficult to read the work
- In general - the resolution of the drawings is too low, which makes them unreadable.
- Figure 1 is poorly legible. It is better to use a block diagram instead of the simplified wiring diagram from the manual.
- The drawings are not very legible, the descriptions are too small, the lack of colors makes reading difficult.
Reviewer 3 Report
See attached file

Round 2
Reviewer 3 Report
The authors have not incorporated my review report. Moreover, they have made more mistakes instead to justify present work. So we reject it.